# Compritol-Based Alprazolam Solid Lipid Nanoparticles for Sustained Release of Alprazolam: Preparation by Hot Melt Encapsulation

**DOI:** 10.3390/molecules27248894

**Published:** 2022-12-14

**Authors:** Huma Rao, Saeed Ahmad, Asadullah Madni, Iqra Rao, Mohammed Ghazwani, Umme Hani, Muhammad Umair, Imtiaz Ahmad, Nadia Rai, Maqsood Ahmed, Kashif ur Rehman Khan

**Affiliations:** 1Department of Pharmaceutical Chemistry, Faculty of Pharmacy, The Islamia University of Bahawalpur, Bahawalpur 63100, Pakistan; 2Department of Pharmaceutics, Faculty of Pharmacy, The Islamia University of Bahawalpur, Bahawalpur 63100, Pakistan; 3Department of Community Medicine, King Edward Medical University Lahore, Lahore 54000, Pakistan; 4Department of Pharmaceutics, College of Pharmacy, King Khalid University, Abha 62529, Saudi Arabia; 5College of Pharmacy, Shenzhen Technology University, Shenzhen 518060, China

**Keywords:** alprazolam, compritol, hot melt encapsulation, solid lipid nanoparticles, sustained release

## Abstract

The current study was designed to investigate the feasibility of incorporating the water-insoluble lipophilic drug Alprazolam (Alp) into solid lipid nanoparticles (SLNs) to offer the combined benefits of the quick onset of action along with the sustained release of the drug. Therefore, compritol-based alprazolam-loaded SLNs (Alp-SLNs) would provide early relief from anxiety and sleep disturbances and long-lasting control of symptoms in patients with depression, thereby enhancing patient compliance. The optimized Alp-SLNs analyzed by DLS and SEM showed consistent particle size of 92.9 nm with PI values and standard deviation of the measurements calculated at <0.3 and negative surface charge. These characteristic values demonstrate the desired level of homogeneity and good physical stability of Alp-SLNs. The SLNs had a good entrapment efficiency (89.4%) and high drug-loading capacity (77.9%). SEM analysis revealed the smooth spherical morphology of the SLNs. The physical condition of alprazolam and absence of interaction among formulation components in Alp-SLNs was confirmed by FTIR and DSC analyses. XRD analysis demonstrated the molecular dispersion of crystalline alprazolam in Alp-SLNs. The in vitro release study implied that the release of Alp from the optimized Alp-SLN formulation was sustained as compared to the Alp drug solution because Alp-SLNs exhibited sustained release of alprazolam over 24 h. Alp-SLNs are a promising candidate to achieve sustained release of the short-acting drug Alp, thereby reducing its dosing frequency and enhancing patient compliance.

## 1. Introduction

During the last few decades, a diverse variety of drug delivery technologies have arisen and the development of nanoscale drug delivery systems constitutes an important feature of this era [1,2]. Solid lipid nanoparticles (SLNs) are an advanced and promising field of nanotechnology with distinct applications in pharmaceutical science [3]. 

The exclusive properties of SLNs that make them potential candidates for drug delivery are smaller size, larger surface area and aqueous and oil phase interactions at the interface [4]. Moreover, the physiochemical diversity of lipids, their compatibility with the biological system, and their potential to enhance bioavailability following oral intake of drugs, have made SLNs very enticing carriers for oral administration [1]. SLNs are least toxic if intended for in vivo applications, as the lipids used to prepare SLNs are usually physiologically compatible and biodegradable with low prospects for acute and chronic toxicity [5]. 

SLNs offer modified or controlled drug release from the lipid matrix, enhance the intracellular uptake of drugs and deliver them at specified rates [6]. They are carriers that transport the entrapped substance to a specific target by crossing various biological barriers, such as BBB (blood-brain barrier). The release of encapsulated drugs in SLNs is controlled by a solid lipid matrix, which also improves the stability of the incorporated ingredients [7].

SLNs are good choice for a lipophilic drug such as alprazolam because decreased mobility of the drug in a solid matrix and improved absorption of this poorly water-soluble drug from SLNs offer potentially beneficial effects. SLNs enhance the solubility and bioavailability of lipophilic drugs and release them far better than emulsions do [8].

SLNs are receiving substantial consideration as promising vehicles to pass the BBB (blood-brain barrier) because they show numerous benefits as compared to other drug delivery systems. For example, SLNs ranging from 120 nm to 200 nm and processed further by hydrophilic surface coating are not easily taken up by the reticuloendothelial system (RES), thereby avoiding liver and spleen filtration. Controlled drug release can be achieved, which may last for several weeks. 

Compritol 888 ATO is known as one of the most valued excipients for the preparation of SLNs. Compritol 888 ATO (glyceryl behenate) is a hydrophobic compound that is a mixture of mono-behenate (12–18%), di-behenate (45–54%) and tri-behenate (28–32%) of glycerol, as shown in Figure 1. It has a melting point range of between 65–77 °C with a hydrophilic-lipophilic balance (HLB) value of 2 [9]. Its drug entrapment is high, also exhibiting its potential in delivering hydrophilic and lipophilic drugs. It is selected by most researchers for SLN formulations because of its favorable characteristics such as non-polar nature and low cytotoxicity compared to other lipids [10]. 

Compritol 888 ATO has been extensively employed to achieve controlled and sustained release of drugs [9]. The recent progress in methods for the synthesis of SLNs, include high-pressure homogenization (HPH), hot melt extrusion coupled with HPH, microchannels, nanoprecipitation using static mixers, and microemulsion-based methods [11]. The current research aimed to synthesize sustained-release SLNs of ALP using a hot melt encapsulation technique. Compritol 888 ATO was selected owing to lipid screening study experiments. Compritol 888 ATO is a novel lipid which is used to attain the modified release of ALP from SLN formulation. 

## 2. Results

### 2.1. Compritol 888 ATO Based ALP-Loaded SLNs Synthesis

#### 2.1.1. Lipid Screening Study

Table 1 presents a list of the lipid screening results. Alprazolam was found to be insoluble in long-chain triglycerides (Dynasan^®^ 114 etc.) and waxes (beeswax). It was soluble to a certain degree in short-chain triglycerides, fatty acids and fatty alcohols. The partial glycerides dissolved ALP to a certain extent, except for Compritol^®^ 888 ATO, which dissolved the highest quantity of drug and showed good solubility potential for SLN formulation development. Moreover, among the partial glycerides, compritol 888 ATO has very good drug entrapment ability due to the presence of long-chain glycerides that improve drug solubilization.

#### 2.1.2. ALP-SLNs

In the current study, water-insoluble drug ALP was successfully incorporated into compritol based SLNs by the HME technique. The effects of lipid and rug ratio on the entrapment efficiency of SLNs and their physicochemical properties were investigated keeping the surfactant concentration constant (1.5%). The ALP content was estimated in SLNs to evaluate various parameters by applying the developed HPLC method [12]. The formulations prepared using various drug and lipid quantities are listed in Table 2.

### 2.2. Entrapment Efficiency (EE) and Drug Loading (DL)

The incorporation efficiency and DL of ALP-SLNs were evaluated by an indirect procedure by assessing the unentrapped quantity of drug from SLNs. The results of these parameters for all formulations are listed in Table 2. Formulation AF6 demonstrated maximum entrapment efficiency (89.4%) and drug loading capacity (77.9%) and was selected as optimized ALP-SLN formulation.

### 2.3. Characterization of ALP-SLNs

The in vitro results of characterization of ALP-SLNs assist in selecting the most suitable formulation composition with desirable in vivo performance. Several formulation variables that influence the quality attributes of SLNs, such as average particle diameter, size distribution, entrapment efficiency, drug loading capacity, zeta surface potential and degree of crystallinity, affect the physical stability, and the extent of drug release from SLNs was evaluated in this study. Various characterization tools and methods have been applied to the current research. The selection of the optimum ALP-SLN formulation was principally based on the minimal particle size, highest entrapment efficiency and maximum drug loading.

#### 2.3.1. Particle Size and Morphological Study

Dynamic light scattering (DLS)

Particle size (PS) is a critical assessment parameter for evaluating the desirable properties of the formulated SLNs. Current research demands the most popular, reliable and quickest method for determining particle size. DLS is the most frequently adopted technique for rapid and accurate assessment of PS and size distribution [13,14]. DLS has various merits such as sensitivity and rapidity, and there is no need for calibration [15]. 

DLS determined average particle size or Z-average (nm) of optimized formulation of ALP-SLNs to be 140 nm. The particle size distribution curve of the optimized formulation is shown in Figure 2. The values of PS of all formulations with zeta potential and polydispersity index (PI) values are tabulated in Table 3.

Scanning Electron Microscopy (SEM)

The morphology of SLNs has a significant effect on the physiological outcomes and efficacy of nanoparticles. SEM can simultaneously determine the particle size, particle shape and surface morphology based on the direct visualization of SLNs [16]. The basic requirement for SEM analysis is the conversion of the SLN formulation to dry powder. The formulation was lyophilized for this purpose.

The surface of SLNs was observed to be spheroid and surface was intact at various magnifications (50, 20, 10, and 5 kX) with sizes ranging from 90 nm to 195 nm. These measurements are consistent with the DLS measurements of the particle size by DLS. Representative SEM photographic images of AF6 are shown in Figure 3. SEM determined average particle size as 144.42 nm using Image J sizer Basics (version 1.38), as shown in Appendix A and Figure 3E.

#### 2.3.2. Polydispersity Index 

PI is frequently measured using DLS instruments. In the current case, PI was observed to be less than 0.5, for all SLN formulations (Table 3) and 0.067 for AF6, indicating uniformity in particle size of the optimized formulation.

### 2.4. Characterization of Surface Chemistry of ALP-SLNs

#### Zeta Potential

The zeta potential curve of the optimized formulation determined by DLS is shown in Figure 4. The Zeta potential values of the compritol-based alprazolam SLNs are negative from −6.71 to −13.81.

### 2.5. FTIR Analysis

FTIR spectrophotometry of the pure drug and formulation components was performed to identify the compatibility between the drug, lipid and surfactant [17]. The FTIR spectrum displayed characteristic peaks corresponding to all the functional groups of the sample. Furthermore, successful incorporation of the drug into SLNs can also be confirmed by FTIR studies of drug, lipid, surfactant, their physical mixture and SLNs [18]. The FTIR spectra of pure drug alprazolam, compritol 888 ATO, Tween 20, physical mixture of drug + lipid (ALP + compritol), and AF6 are shown in Figure 5. 

The FTIR spectra of alprazolam Figure 5A revealed principal characteristic peaks of C-Cl stretch at 694.35 cm^−1^, aromatic C-H stretch at 3053.12 cm^−1^, phenyl group peak at 1608.47 cm^−1^, C-H stretching at 1485.64 cm^−1^, aromatic C-H bending at 777.43 cm^−1^. The characteristic peaks of C-N and C=N groups of azomethine ring of alprazolam appeared at 1312.47 cm^−1^ and 1608.47 cm^−1^, respectively. 

The FTIR scan in Figure 5B of compritol 888 ATO showed a broad vibrational band due to the -OH stretching between 3100 and 3450 cm^−1^. Several vibrational bands appeared in the area between 700 and 1450 cm^−1^ showing the presence of methylene groups of the long chain of triglyceride. The characteristic peak of the carbonyl C=O stretch appeared at 1735 cm^−1^ and the absorption bands of C-H stretching appeared at 2847 and 2914 cm^−1^. The appearance of a strong peak at 1175 cm^−1^ is a characteristic feature of the C-H bending vibration.

The FTIR spectrum of Tween 20 in Figure 5C showed a wide OH band at 3470, and absorption bands of the C-H stretch of the aliphatic alkyl group appeared at 2853 and 2920 cm^−1^ in Figure 6C. The C=O stretch appeared as a peak at 1734 cm^−1^ and a strong signal of C=O in plane bending appeared at 719.50 cm^−1^. The peaks observed at 1415 cm^−1^ are due to the bending vibration of the -OH group. A strong stretch of the C-O-C characteristic of Tween 20 appeared at 1095 cm^−1^ and 948 cm^−1^. 

The FTIR scan of the physical mixture (ALP-compritol) showed characteristic peaks of alprazolam pure drug, as well as compritol, without any distinct shift. The absorption bands of the C-H stretch of the aliphatic alkyl group appeared at 2853 and 2920 cm^−1^. Several vibrational bands appeared in the area between 700 and 1450 cm^−1^ showing the presence of methylene groups of the long chain of triglyceride. The C-N and C=N groups of the azomethine ring of alprazolam also appeared in the physical mixture spectrum. This showed the absence of physical or chemical interactions between the drug and lipid in SLNs. 

The distinct characteristic peaks of alprazolam and compritol were broadened in the FTIR spectrum of the AF6 (Figure 5E). This might be due to the overlapping of the drug peaks with other formulation constituents. The broadening of characteristic peaks in AF6 proves that, in SLNs, the lipid forms the external core, with the drug successfully incorporated internally in this core [18].

### 2.6. Differential Scanning Calorimetric (DSC) Analysis

DSC is a reliable and rapid thermo-analytical technique that pinpoints the nature of the sample, either crystalline or amorphous, along with the degree of crystallinity of lipids in SLNs and can determine the enthalpy change to evaluate the drug incorporation into nanoparticles, and investigate interactions between drugs and nanocarriers in SLNs [15]. 

DSC analysis of alprazolam, compritol 888 ATO, alprazolam-compritol ATO 888 physical mixture and lyophilized AF6 was performed and results are presented in Figure 6. 

### 2.7. X-ray Diffraction Analysis

Powder XRD patterns were recorded in the form of diffractograms. During the characterization of ALP-SLNs, X-ray diffraction analysis of ALP, compritol, and AF6 was performed to check the number of crystalline peaks and their intensity in order to observe the formation of the drug-lipid matrix in SLNs, as shown in Figure 7. 

### 2.8. In Vitro Drug Release Study

The cumulative percentage drug release from ALP-SLNs was determined and was found to show ALP release in a sustained manner over a duration of 30 h (Figure 8). 

### 2.9. Calculation of Release Kinetics

Evaluation of drug incorporation and release is a prominent tool for designing promising drug carrier systems. The release kinetics of the compritol-based ALP-SLNs were investigated using different kinetic models. The results of curve fitting into these models are presented in Table 4. 

## 3. Discussion

Alprazolam is a short-acting drug which requires repeated dosing to achieve and maintain constant plasma profiles. In order to avoid frequent dosing, it is desirable to achieve sustained release of the alprazolam to reduced dosing frequency and enhance patient compliance. The prepared ALP SLNs will improve the ability of drugs to reach the CNS by crossing the BBB due to their lipophilicity, smaller size and larger surface area, requiring less dose. Compritol 888 ATO is superior to homogenous glycerides owing to the presence of a large amount of mono-, di-, and triglycerides, which assists in better drug solubilization and imparts better drug entrapment ability to compritol [19,20].

The physicochemical features of SLNs, such as entrapment efficiency (EE) and drug loading (DL), depend upon the solubility of the drug in lipid media [21,22]. It is anticipated that SLNs demonstrate a higher loading capacity and encapsulation efficiency if the drug solubilizes in the melted lipid as compritol does, having a lower melting point, in contrast to alprazolam, when incorporated into SLNs [23]. 

An increase in EE was observed with increasing concentration of compritol 888 ATO (5:1 to 30:1) in the formulations. Compritol 888 ATO leads to higher drug entrapment, as it is a mixture of mono-, di-, and triglycerides, and forms a less ordered arrangement of lipid crystals. This imperfection, or lattice defects in compritol, provide more loading space for the accommodation of the drug [24]. Increasing the lipid concentration from 30:1 to 50:1 resulted in a reduction in the entrapment of ALP in SLNs. The reduced entrapment suggests that, when SLNs have a drug to lipid ratio above a certain level, the crystallization phenomenon of the lipid phase induces an incomplete expulsion of alprazolam on the particle surface. Moreover, a higher lipid concentration generates an interface with high viscosity that will lead to reduced diffusion of the solvent and, consequently, only some lipid molecules are transported to the aqueous phase. Thus, at these higher concentrations, the formation and stabilization of lipid aggregates receded [25]. 

AF6 SLNs showed high entrapment efficiency (89.4%) for alprazolam as compared to GMS based alprazolam loaded SLNs for nose to brain delivery 40.32 ± 0.5% [26]. For transdermal delivery of alprazolam, its nanoliposomes were prepared through encapsulation of alprazolam in nanoliposomes using ethanol injection [27]. Alprazolam-loaded chitosan-egg albumin-PEG polymeric nanoparticles were prepared using a heat coagulation method to achieve sustained release [28]. 

SEM images of the ALP-SLNs revealed the uniform and spherical shape of the particles. The particle size of optimized formulation 90.92 nm measured by SEM was consistent with DSL measurements.

The negative zeta potential (−6.71 mV) of AF6 implies the prominently stable dispersion characteristic of SLN formulations. This negative zeta potential provides sufficient stability and shelf life for SLNs and makes nanoparticles less prone to aggregation [18,29]. GMS based ALP SLNs showed a negative charge of −11.2 ± 0.034 mV [26].

DSC thermogram of ALP showed a sharp endothermic peak at 227 °C corresponding to the melting of ALP. This endothermic peak starts at the onset point of 222 °C and ends at 234 °C. The DSC scan of compritol presented a sharp internal peak at its melting point (72.89 °C), confirming the crystalline nature of compritol. The thermogram of the drug (alprazolam)-lipid (compritol 888 ATO) mixture showed a lower calorimetric peak at a slightly lower temperature of approximately 221 °C than the original endothermic peak of ALP. DSC scan of lyophilized ALP SLNs did not exhibit a sharp lower calorimetric peak at the melting point of alprazolam, as observed in the DSC scans of the pure drug and physical mixture. There is a peak at about 140 °C, showing the least partly crystalline state of ALP, and not fully amorphous. The disappearance of the peak demonstrated complete solubilization and uniform dispersion of alprazolam in the molten compritol matrix and an amorphous state in the alprazolam-SLNs [30].

The melting endothermic peak of compritol shifted slightly to a lower temperature (69.6 °C) in AF6. The decrease in melting point in SLNs compared with bulk lipid was attributed to the decrease in particle size associated with high specific surface area, dispersibility of lipids, and presence of water and surfactants [30]. Moreover, the reduced values of melting enthalpy indicate less ordered latticework arrangement of compritol in AF6 contrary to the bulk state of lipid [24,30]. 

X-ray diffractograms for ALP and compritol exhibited sharp crystalline peaks owing to the crystalline nature of the drug and lipid. These sharp peaks are absent in the diffractogram of alprazolam SLNs, indicating molecular dispersion of alprazolam in compritol in AF6. This can also demonstrate dispersion at the molecular level with a decrease in the degree of crystallinity [15].

The drug-release behavior was studied simulating physiological conditions using simulated intestinal fluid (pH 6.8 and 7.4), as pH of GIT varies from small intestine to ileum.

A comparative study involving in vitro release at pH 6.8 and 7.4 was performed using alprazolam solution (ALP-Sol) and AF6. The release data profile at pH 6.8 and 7.4 from AF6 exhibited a bimodal release pattern comprising the initial burst release of ALP pursued by slow sustained release. AF6 showed an initial release of 15.809% after 1 h, whereas the drug solution showed 42.578% after 1 h. AF6 demonstrated sustained release of ALP up to a maximal release of 85.12% in 30 h, whereas the drug solution exhibited 98.16% release within 7 h. The initial release of the drug might be due to the presence of the free drug in the external phase or weakly bound drug on the SLN surface, whereas the drug incorporated in the lipid core of AF6 is released in a sustained manner [17]. 

The degree of burst release from SLNs is controlled by controlling the extent of drug solubility in the aqueous surfactant phase during SLN production, which is dependent on the surfactant concentration employed and temperature used for SLN synthesis. Higher concentrations of surfactants and higher temperatures increase the burst release, while the production of SLNs at room temperature prevents drug partitioning into the aqueous phase and successive re-partitioning of the drug into the lipid phase, hence exhibiting no burst release of the drug at all. In AF6, the burst release of the drug was controlled by using compritol that completely dissolved the drug and the temperature used for SLN synthesis was from 85 °C to room temperature, thereby reducing abrupt burst release and providing sustained release of ALP. 

In vitro drug release kinetics comparatively analyzed the values of R^2^ for various models and results revealed that the Higuchi model was the best fit model for ALP-SLNs. The order of the R^2^ values obtained was as follows: Higuchi model 0.9880 followed by Korsmeyer-Peppas model (R^2^ = 0.9687), Weibull (R^2^ = 0.9590), first-order (R^2^ = 0.9197), and zero-order (R^2^ = 0.8542). 

Korsmeyer-Peppas model is expressed as
Mt/M∞ = Kt^n^(1)
where Mt/M∞ is the fractional drug release at time t, “Mt” is the mass (amount) of ALP released at time t, “M∞” is the total mass (amount) of ALP in dosage form, “K” shows kinetic constant which is specific for the drug/lipid system, and “n” is the diffusion or release exponent. Its value indicates the release mechanism of the drug from the formulation. 

The value of the release exponent (n) for Korsmeyer-Peppas for optimized ALP-SLNs was 0.644, indicating an anomalous transport mechanism pursued by ALP for release from AF6. The anomalous transport mechanism releases drugs by non-Fickian diffusion, which demonstrates both diffusion-controlled and erosion-controlled release of drugs from AF6, leading to sustained drug release [31]. 

## 4. Materials and Methods

### 4.1. Materials

Alprazolam was received as a gift from Genix Pharmaceuticals (Pvt) Ltd. Karachi, Pakistan. Percirol ATO 5 and Compritol^®^ 888 ATO were procured from Gattefosse, 36 Chemin de Genas, Saint-Priest Cedex, France. Tween 20 was kindly supplied by Daejung Chemicals & Metals, Siheung-si, Republic of Korea. Dynasan 114, Dynasan 116, Dynasan 118, and Witepsol E 85 were obtained from IOI Oleo GmbH Hamburg, Germany. All other reagents and chemicals used were of analytical grade. HPLC-grade deionized water was used for all experiments. 

### 4.2. Lipid Screening Study

The fundamental step in SLN formulation development is the lipid screening study, which includes the observation of drug solubility in various solid lipids. For this purpose, the lipids were melted in a 20 mL glass vial, the drug was added, followed by stirring of the mixture at 125 RPM at 80 °C for 1 h. The resultant drug-lipid mixture was allowed to cool and its solubility was checked afterward. The solubility behavior was evaluated after solidification of the molten lipid, both visually and by light microscopy. During solidification, the vials were manually rotated slowly to enable a thin layer of the mixture to solidify on the walls of the glass vial, which was observed for solubility. Furthermore, in light microscopy, the drop of the molten commixture was taken on a microscope slide covered with a cover glass and compressed to form a thin film. After waiting for a while to allow solidification of the film, the slide was observed by light microscopy to check for the presence of drug crystals.

The solubility of alprazolam and desvenlafaxine was higher in compritol than in all studied lipids. Hence, “Compritol^®^ 888” ATO was selected as the lipid phase for the SLNs. After selection of the formulation components, the quantity of the components was optimized by choosing the appropriate proportions of lipids, surfactants and drugs. 

### 4.3. Preparation of Alprazolam-SLNs

The Alprazolam-loaded-SLNs were prepared by using HME technique followed by homogenization. Alprazolam (1 mg) was weighed accurately and then added to an appropriate amount of Compritol 888 ATO. The mixture of alprazolam and compritol was melted at 82 °C using a magnetic stirrer by Velp Scientifica, Deer Park, NY, USA. Tween 20 was separately dissolved in deionized water and heated to the same temperature, i.e., 82 °C. When the lipid melts a clear homogenous lipid phase, a mixture of drugs and lipids is formed. The hot aqueous surfactant solution was added at 500 RPM to the hot lipid phase and maintained at 82 °C. The stirring speed was increased to 700 RPM maintaining the temperature at 82 °C for 6 min. Subsequently, heating was stopped and stirring was continued for 90 min until the formulation reached room temperature. The resultant formulation was homogenized at 12,000 RPM for 3 min using a high-speed homogenizer, Heidolph Silent Crusher M (Alberta, AB, Canada). The obtained formulation was sonicated for 15 min using an Elmasonic sonicator (E 30 H, Singen, Germany). The process of preparing the alprazolam-loaded SLNs is shown in Figure 9.

### 4.4. Particle Size and Morphological Study

Particle size (PS) is a crucial parameter for evaluating the physicochemical properties of SLNs. It is highly recommended that PS be determined by more than one complementary technique to enhance the reliability of the results [32]. DLS and SEM were used to determine particle size in the current study. 

#### 4.4.1. Dynamic Light Scattering (DLS)

The particle size (PS) and polydispersity index (PI) of alprazolam-loaded SLNs were measured by using DLS on a Zeta sizer (Nano ZS, Malvern Instruments, Malvern, UK). Each formulation was dispersed in double-distilled water, followed by ultra-sonication for 5 min before size measurement. The homogeneous SLN dispersion was analyzed for the PS and PI. Each sample was analyzed in triplicate and the average values were considered.

#### 4.4.2. Scanning Electron Microscopy (SEM)

SEM analysis of SLNs was performed by utilizing SEM-JEOL, Tokyo, Japan with working parameters of voltage (7–10 kV) and magnifications (50, 25, 10, 5) kX. Samples were prepared on a carbon stub and gold-coated for 3 s using a sputter coater prior to SEM analysis. 

### 4.5. Surface Chemistry of Alprazolam SLNs

#### Zeta Potential 

The surface charge of the SLNs was indirectly estimated by measuring the zeta potential of the ALP SLN formulation. Zeta potential values of the same homogenous suspension prepared for PS and PDI analysis were determined by using a Zeta sizer (Nano ZS, Malvern Instruments, Malvern, UK) in a replaceable folded capillary zeta cell at 25 °C. For statistical evaluation, each sample was analyzed in triplicate, and the mean of the values along with the standard deviation of the measurements were calculated.

### 4.6. Entrapment Efficiency (EE) and Drug Loading (DL)

The dialysis method was employed to determine the EE of alprazolam in the SLNs after separating the unentrapped drug [33]. 2 mL of the drug-loaded SLN formulation was placed into a dialysis bag (MW cutoff 8 kDa, Float-A-Lyser^®^. G2, Sigma, St. Louis, MO, USA). The dialysis bag was immersed in a glass tube containing 10 mL of phosphate buffer saline (PBS) and stirred at 400 RPM for 30 min. The dialysate was then filtered through a nylon syringe filter 0.45 µm. The quantity of alprazolam in the dialysate was determined using HPLC [12]. After calculating the quantity of free ALP in the buffer, the drug EE in the SLNs was measured using the Equation (2):(2)$$EE\;\%=\left(\frac{Wi-Wd}{Wi}\;\right)\times 100$$
where *Wi* is the weight of the drug initially used in the formulation, and *Wd* is the drug quantity which is measured in the dialysate.

The percent drug loading (DL%) of ALP-SLNs was calculated using the following Equation (3):(3)$$DL\;\%=\left(\frac{Wi-Wd}{Wi+WL}\;\right)\times 100$$
where *Wi* is the weight of the drug initially used in the formulation, Wd is the drug quantity which is measured in the dialysate after dialysis and *WL* is the weight of the lipid used in the formulation.

### 4.7. HPLC Analysis of Drug

HPLC is an efficient technique for evaluating the efficiency of nano-drug formulations, including SLNs. The HPLC method was used to analyze alprazolam to investigate drug loading, entrapment efficiency and alprazolam in vitro release in the SLN system. The instrument used was HPLC (Perkin Elmer series 200, MA, USA). The column utilized was a reversed-phase hypersil C18 BDS column (250 mm × 4.6 mm, 5 μm). The mobile phase used for analysis was a combination of acetonitrile and 0.02 M KH_2_PO_4_ buffer (65:35 *v*/*v*) with 0.1% trifluoroacetic acid (TFA). A 1 M potassium hydroxide solution was used to reconcile the pH of the mobile phase to 4.00. The flow rate of the mobile phase was 1.00 mLmin^−1^ and the injection volume was 20 μL. Quantitation was based on peak area measurements at 230 nm [12]. The retention time of alprazolam was 5.182 min. 

### 4.8. Fourier Transform-Infrared (FTIR) Spectrophotometric Analysis

FTIR spectra of alprazolam, compritol 888 ATO, surfactant and a physical mixture of the components, and prepared SLN formulation (ALP-SLNs) were obtained using an FTIR spectroscope (Perkin Elmer Spectrum RX I, MA, USA). The samples were reduced to powder and then analyzed in the form of KBr pellets by placing the pellet in the sample holder of the instrument. The wavelength range from 4000 to 400 cm^−1^ was used for spectral scanning at 4 cm^−1^ resolution, with 2 mms^−1^ scan speed. 

### 4.9. DSC Analysis

The DSC analysis of alprazolam, compritol 888 ATO and lyophilized optimized ALP-SLNs was performed using a differential scanning calorimeter (DSC) Q2000 (Pyris 6 DSC Perkin Elmer, Shelton, CT, USA) in an inert nitrogen atmosphere. 10 milligrams of samples were sealed on a pan made of aluminum, and an empty pan of aluminum served as a reference. The samples were heated at a rate of 10 °C/min over a temperature ranging 40–300 °C. The thermal behaviors of the samples were observed under a nitrogen purge and DSC scans or thermograms were recorded. 

### 4.10. XRD Analysis

DSC is combined with other techniques such as microscopy, TGA and XRD analysis for the characterization of SLNs. X-ray diffraction measurements were performed by using an X-ray diffractometer (JDX-3532, JEOL, Tokyo, Japan) at 20–40 kV. X-rays of Cu-Ka (λ = 1.5418Å) radiation source were utilized at a scanning rate of 2 h/min at 5 °C per min with 2Theta-Range: 0 to 160°. The measurements were carried out on bulk drug ALP, bulk lipid compritol and AF6.

### 4.11. In-Vitro Release Study

The prepared nanoparticles and standard ALP solutions were evaluated for in vitro drug release by using the dialysis bag diffusion technique. 5 mL AF6 was taken in a dialysis bag (m.w. cutoff 12,000 Da, Biotech Grade membranes, Shanghai, China) prepared after tie up of the two ends of bag by a thread. The bag was suspended in dissolution media (500 mL phosphate buffer saline pH 6.8) preheated to 37 °C in a USP type II dissolution apparatus. The stirring speed of the medium was set to 50 RPM. The dialysis bag containing the SLN formulation was a donor compartment, and the vessel containing dissolution media served as the receptor compartment. An aliquot of 3 mL was withdrawn at predetermined time intervals (0, 0.25, 0.5, 0.75, 1, 2, 4, 6, 9, 12, 16, 20, 24, 30, 36, 42, and 48 h) from the dissolution vessel and replaced with a similar volume of fresh medium maintained at the same temperature i.e., 37 °C. The withdrawn aliquots were filtered using a 0.45 µm nylon syringe filter and analyzed by the aforementioned HPLC method to evaluate the percentage of drug release from SLNs. All experiments were performed in triplicate. An in vitro drug release study was performed in phosphate buffer at pH 7.4 by repeating the same procedure.

### 4.12. In-Vitro Drug Release Kinetics 

In order to evaluate the release pattern of alprazolam from AF6, the set of data attained from in vitro release study was subjected to kinetic analysis with mathematical models such as zero order, first order, Higuchi release model, Korsmeyer-Peppas, and the Weibull model by in vitro dissolution kinetic modelling DD Solver. The (R^2^) for each of these models were calculated for each formulation. The results of the kinetic models predicted the release of ALP from AF6 and mechanism of drug transport.

## 5. Conclusions

In the current study, alprazolam-loaded compritol based SLNs were successfully synthesized by utilizing the HME technique by optimizing formulation components and their ratio. HME seemed to be a good technique for attaining high entrapment efficiency and drug loading in AF6. HPLC techniques was used for characterization of ALP-SLNs instead of a traditional UV spectrophotometer. The results of physicochemical characterization of AF6 exhibited small particle size, narrow PDI, and a high entrapment efficiency which highlights that AF6AF6, having desirable dispersion stability and long circulation time, would keep ALP at the site of action, and prolonged antianxiety effect may be achieved by these ALP-SLNs. An in vitro release study implied sustained release of alprazolam from AF6 formulation over 24 h. Alprazolam is a habit-forming drug; a smaller dose is considered to be more suitable for a patient so its sustained release would provide greater therapeutic outcomes and improve patient compliance. Overall, the aforementioned results depicted the potential of these compritol-based ALP-SLNs for sustained release of the lipophilic drug ALP.

## Figures and Tables

**Figure 1 molecules-27-08894-f001:**
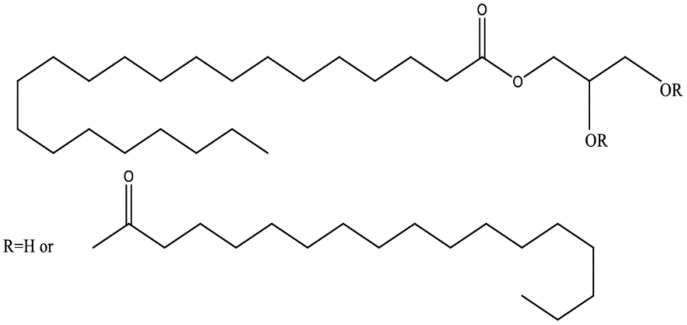
Chemical structure of Compritol 888 ATO.

**Figure 2 molecules-27-08894-f002:**
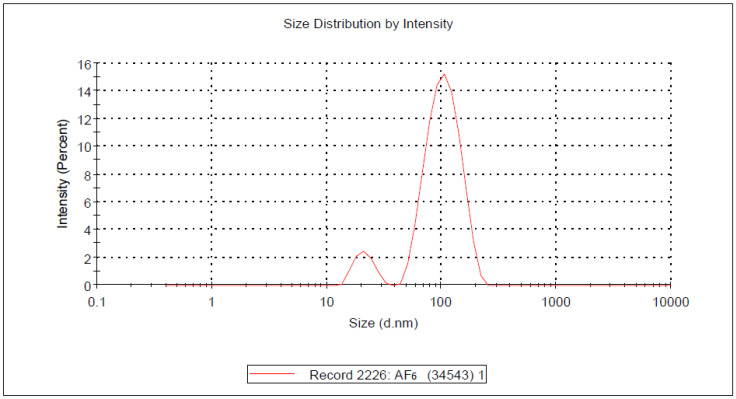
Particle size distribution curve of AF6.

**Figure 3 molecules-27-08894-f003:**
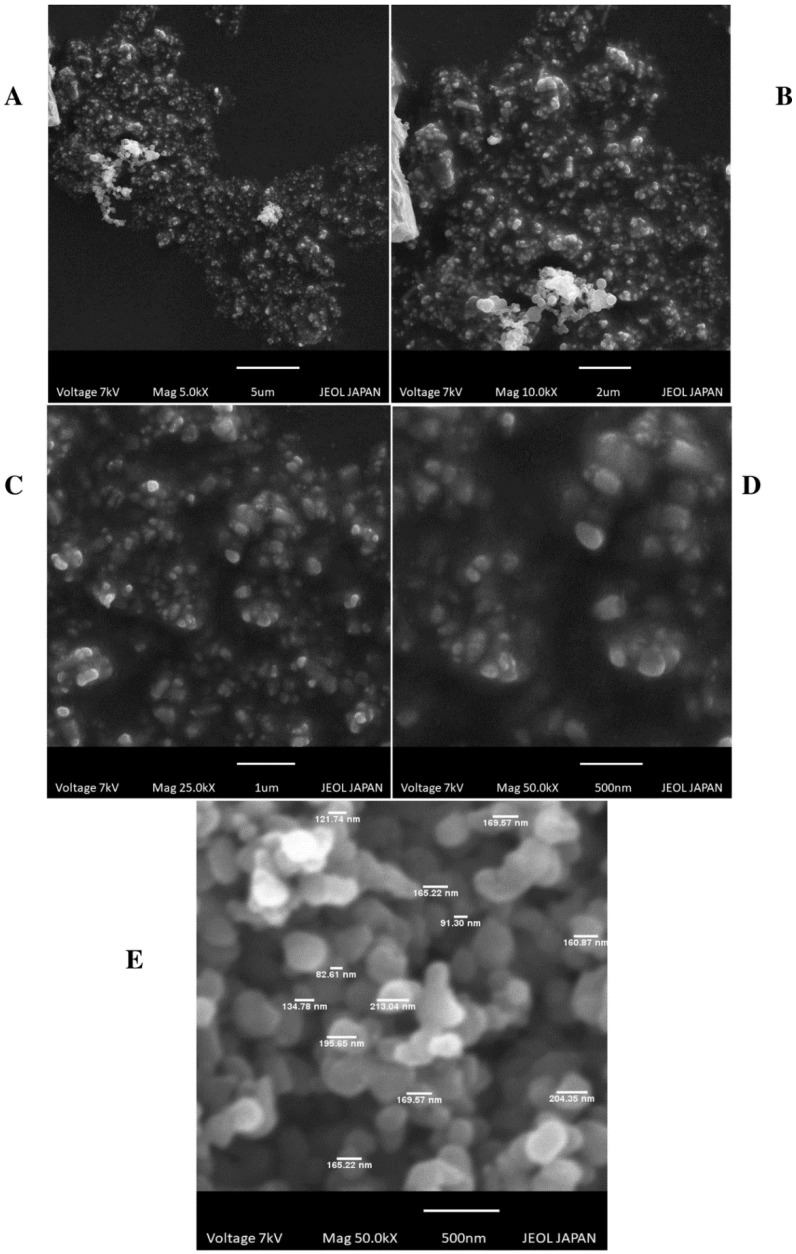
SEM images of AF6 at magnification (**A**) 5 kX (**B**) 10 kX (**C**) 25 kX (**D**) 50 kX (**E**) Individual particle size determination of ALP-SLNs by Image J sizer.

**Figure 4 molecules-27-08894-f004:**
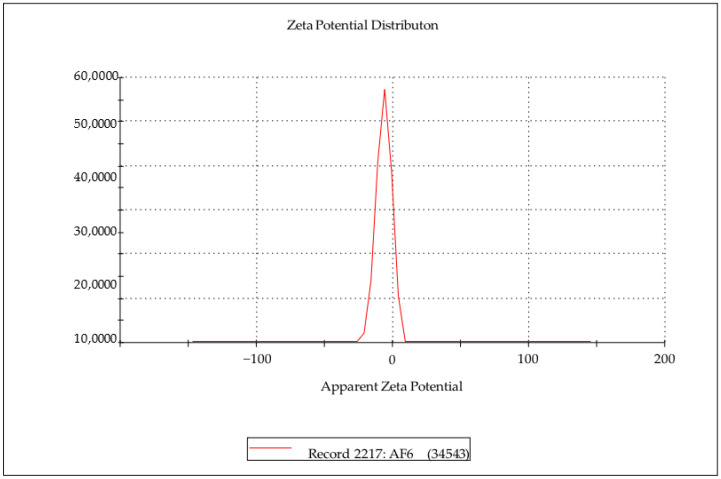
Zeta potential distribution curve of AF6.

**Figure 5 molecules-27-08894-f005:**
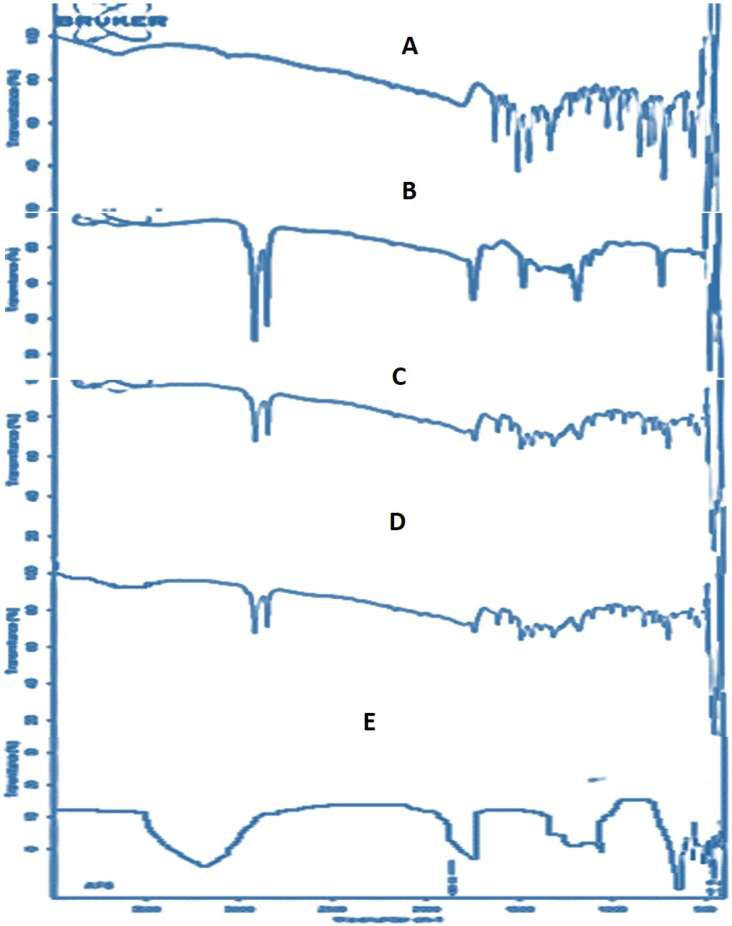
FTIR spectrum of (**A**) pure alprazolam (**B**) compritol 888 ATO (**C**) Tween 20 (**D**) physical mixture (ALP + compritol) (**E**) AF6.

**Figure 6 molecules-27-08894-f006:**
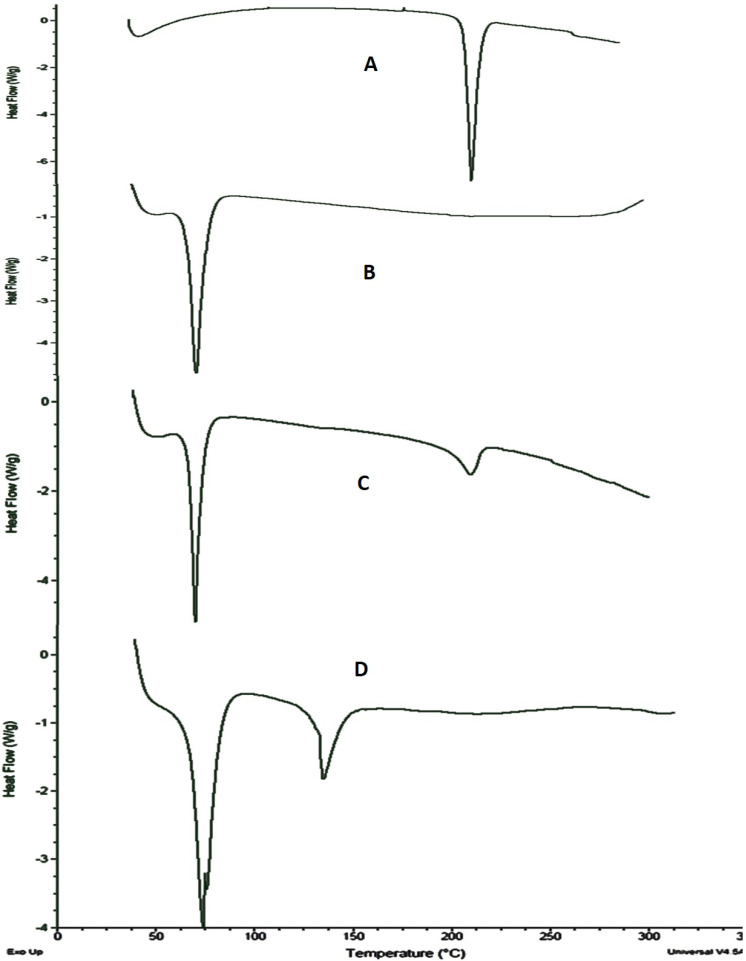
DSC thermogram of (**A**) alprazolam (**B**) compritol 888 ATO (**C**) physical mixture of alprazolam+ compritol 888 ATO (**D**) AF6.

**Figure 7 molecules-27-08894-f007:**
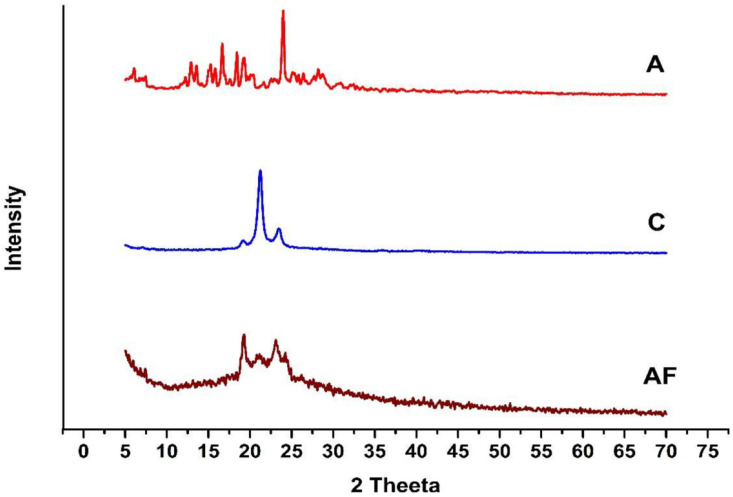
X-ray diffractogram for A (alprazolam), C (compritol), and AF6.

**Figure 8 molecules-27-08894-f008:**
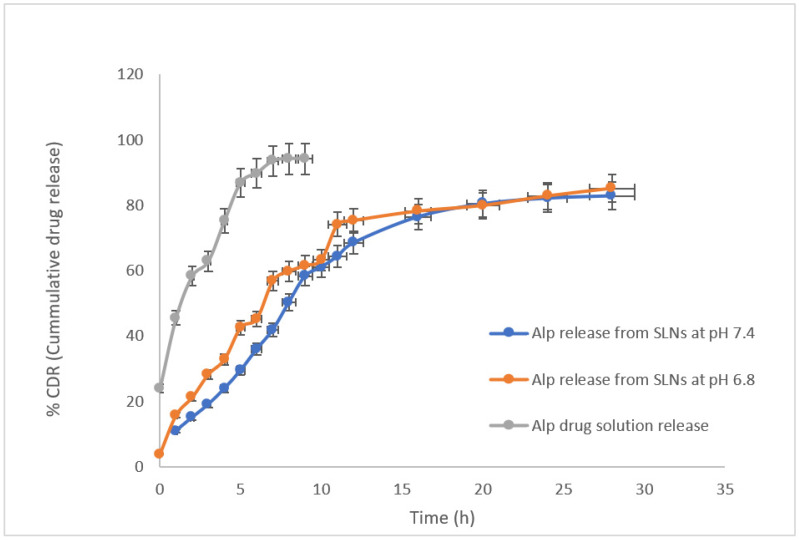
In vitro release profile of ALP drug solution & AF6.

**Figure 9 molecules-27-08894-f009:**
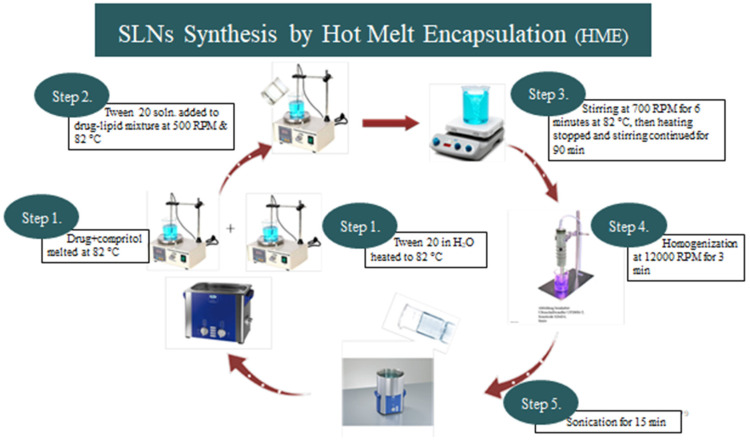
HME technique for preparation of compritol based alprazolam SLNs.

**Table 1 molecules-27-08894-t001:** Solubility of Alprazolam in various lipids.

Lipids	Time (Minutes)			
	15	30	45	60
Waxes				
Beeswax	−	−	−	−
Monoglycerides				
Glyceryl caprylate (Imwitor^®^ 308)	−	−	−	−
Glyceryl laurate (Imwitor^®^ 312)	−	−	−	−
Glyceryl stearate (Imwitor^®^ 900 P)	−	−	−	+
Triglycerides				
Trimyristin (Dynasan^®^ 114)	−	−	−	−
Tripalmitin (Dynasan^®^ 116)	−	−	−	−
Tristearin (Dynasan^®^ 118)	−	−	−	−
Diglycerides				
Glyceryl palmitostearate (Precirol^®^ ATO)	−	−	+	+
Glyceryl behenate (Compritol^®^ 888 ATO)	+	+	+	+
Fatty acids				
Oleic acid	−	−	+	+
Stearic acid	−	−	−	+

(+): soluble; (−): insoluble.

**Table 2 molecules-27-08894-t002:** Formulation chart for alprazolam loaded SLNs with Entrapment efficiency and Drug loading.

ALP-SLNs	Lipid: ALP	Lipid: Surfactant	Entrapment Efficiency (%)	Drug Loading	Drug Loading (%)
AF1	5:1	1:3	42.35	1.031	51.6
AF2	10:1	1:3	54.68	1.333	66.65
AF3	15:1	1:3	57.75	1.408	70.4
AF4	20:1	1:3	74.75	1.823	71.15
AF5	25:1	1:3	78.65	1.868	73.4
**AF6**	**30:1**	**1:3**	**89.4**	**1.958**	**77.9**
AF7	35:1	1:3	63.45	1.547	72.35
AF8	40:1	1:3	67.8	1.653	62.65
AF9	45:1	1:3	63.55	1.793	59.65
AF10	50:1	1:3	64.35	1.663	53.15

**Table 3 molecules-27-08894-t003:** Mean particle size (nm), zeta potential (mV), and polydispersity index (PI) of different ALP formulations.

Formulation Code	Particle Size (nm)	Zeta Potential (mV)	PI
AF1	100.72 ± 036	−6.87 ± 0.25	0.362 ± 0.12
AF2	117.39 ± 0.0	−8.17 ± 0.15	0.443 ± 0.23
AF3	127.39 ± 0.49	−9.37 ± 0.15	0.415 ± 0.26
AF4	104.35 ± 0.52	−7.36 ± 0.125	0.175 ± 0.14
AF5	126.09 ± 0.58	−8.98 ± 0.25	0.359 ± 0.31
AF6	140.34 ± 1.13	−6.71 ± 0.126	0.067 ± 0.08
AF7	180.36 ± 1.18	−13.81 ± 0.35	0.312 ± 0.05
AF8	147.83 ± 1.0	−11.6 ± 0.05	0.135 ± 0.31
AF9	155.65 ± 1.0	−13.5 ± 0.1	0.247 ± 0.41
AF10	130.43 ± 1.5	−8.39 ± 0.10	0.429 ± 0.06

**Table 4 molecules-27-08894-t004:** Release kinetic models for AF6.

Optimized ALP-SLNs	Correlation Coefficient (R^2^)					Release Exponent (n) of Korsmeyer-Peppas
	Zero-Order	First Order	Higuchi	Korsmeyer-Peppas	Weibull Model	
AF6	0.8542	0.9197	0.9880	0.9687	0.9590	0.644

## Data Availability

All data available in the manuscript.

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
