# Peer review of "Compritol-Based Alprazolam Solid Lipid Nanoparticles for Sustained Release of Alprazolam: Preparation by Hot Melt Encapsulation"

_molecules, 2022, doi:10.3390/molecules27248894_

Round 1
Reviewer 1 Report
In this manuscript, the authors incorporated the water-insoluble lipophilic drug alprazolam into solid lipid nanoparticles in order to provide a comprehensive benefit of rapid onset and sustained drug release. This work could be considered for publication after major revision. Other questions are shown below.
1. The format of references is not uniform, such as [20] [21] and [22, 23]. Consult the journal's reference style for the exact appearance of these elements, and use of punctuation and capitalization.
2. Graphical abstracts miss punctuation, such as in Figure 6. The format of icons should also be as beautiful as possible.
3. The structure of the article needs to be further improved. The structure of the methods part is too compact.
4. Abbreviations that have been explained do not need to be explained later, for example, hot melt encapsulation (HME) appears many times in the article.
5.Check all format, Mt/M∞ = Ktn in 492, t should be in subscript, check all.
6. ln the conclusions, in addition to summarizing the actions taken and results, please strengthen the explanation of their significance. It is recommended to use quantitative reasoning comparing with appropriate benchmarks, especially those stemming from previous work.
7. The figures are not quite professional.
Author Response
Reviewer 1
In this manuscript, the authors incorporated the water-insoluble lipophilic drug alprazolam into solid lipid nanoparticles in order to provide a comprehensive benefit of rapid onset and sustained drug release. This work could be considered for publication after major revision. Other questions are shown below.
Response: The authors are highly thankful to the reviewer for sparing his quality time and generating useful comments. These comments were very helpful in improving our manuscript and will also be considered in our future research. All these comments have been addressed point by point and highlighted in Green colour.
- The format of references is not uniform, such as [20] [21] and [22, 23]. Consult the journal's reference style for the exact appearance of these elements, and use of punctuation and capitalization.
Response: The reference section was formatted according to journal's reference style.
- Graphical abstracts miss punctuation, such as in Figure 6. The format of icons should also be as beautiful as possible.
Response: The format of icons has been improved.
- The structure of the article needs to be further improved. The structure of the methods part is too compact.
Response:
- Abbreviations that have been explained do not need to be explained later, for example, hot melt encapsulation (HME) appears many times in the article.
Response: Abbreviation was replaced.
- Check all format, Mt/M∞ = Ktn in 492, t should be in subscript, check all.
Response: The format has been checked and corrected.
- ln the conclusions, in addition to summarizing the actions taken and results, please strengthen the explanation of their significance. It is recommended to use quantitative reasoning comparing with appropriate benchmarks, especially those stemming from previous work.
Response: Thank you for your valuable suggestion. Conclusion section has been revised as per your comment.
- The figures are not quite professional.
Response: The figures have been improved professionally.
Reviewer 2 Report
The authors of the manuscript present the synthesis and characterization of drug Alprazolam /solid lipid nanoparticle suspension using hot melt encapsulation technology. The influence of formulation parameters was investigated. The study is interesting and contributes to scientific relevance to the theme. The cited references give scientific support to the proposed study. The methodology consists of several studies, well-founded, and the results and discussion are presented in most cases clearly. The present work is good in quality, but I recommend the publication after a major revision, as follows:
Line 22: “PI values < 0.3 nm” PI has no unit
Introduction: there is a luck of previous studies about alprazolam SLS composites (although it is mentioned in section 4 in lines 433-442 (27. Singh, A.P., S.K. Saraf, and S.A. Saraf, SLN approach for nose-to-brain delivery of alprazolam. Drug delivery and translational research, 2012. 2(6): p. 498-507.)
Introduction: Please, mention possible preparation methods for SLS composites. E.g. from Khairnar et al. Review on the Scale-Up Methods for the Preparation of Solid Lipid Nanoparticles. Pharmaceutics 2022, 14,1886. https://doi.org/10.3390/pharmaceutics14091886 or others. The author make it clear in the main part that the ALP-SLNs are suspension in form.
In line 82: when the „current research aimed to synthesize SLNs of ALP using Compritol 888 ATO” why was lipid screening done? (Section 2.2. and 3.1.1.). If the authors want to insert Section 3.1.1. in this paper some information must be provided about possible other lipids and mentioned the results in Discussion.
In line 152 „…along with the standard deviation of the measurements were calculated”. Standard deviation data are not given in Table 3.
Section 2.9. What was the reason of the selected two pH for the drug release studies? Please, explain it in Discussion.
Section 2.11 and 2.12. : Please, put after Section 2.8 because these methods are also for physical characteristics determination (similar with Section 3.8 and 3.9.)
In line 226: the XRD investigation was made of “dried alprazolam formulation (ALP-SLNs)”, please, complemented it.
Table 2: Because of the used surfactant concentration and the lipid:surfactant ratio were constant there is no need to put in the table. The authors can give them as non variable factors in Section 2.3. What is the drug loading without % (ratio?)?
In line 257: please correct it: “parameters for all formulations are listed in Table 2”.
In line 292: “The shape of SLNs was observed to be spheroid and the surface intact at various magnifications” please, correct it. Generally comment: please check where is more ALP-SLNs and where is only one sample (e. g. AF6, is it in Fig.4 as SEM or FTIR, XRD, DSC). Can be signed as AF6 and not ALP-SLNs in figures and also in the text.
In line 298, Table 4: it is not necessary to give all the detailed data of PS by SEM, the author can give only the average value in text. The Table 4. can be omitted (detailed data can be put in supplementary materials).
In line 310: “The Zeta potential values of the compritol-based alprazolam SLNs are highly negative 310 from −6.71 to −13.81.” The colloid dispersion can be considered stable if the zeta potential higher than ± 30 mv (e. g. Kashanian et al. https://doi.org/10.2147/IJN.S20849). But it should be noted that it is especially for electrostatic stabilized colloids. The “highly” is not appropriate for these values. Similar in line 438.
Line 329: In Fig. 6. E is an absorbance spectrum?
In line 362: “…show ALP release in a sustained manner over a duration of 30 h” please, completed.
In line 411: Please, give the BBB meaning at first appearance
In lines 420-425: The amount of the lipid and ALP were given in ratio in Table 2, there is no data of 10 mg and 80 mg and %. E. g. AF1 has 5 mg lipid and 1 mg ALP. Better to give the ratio or the sample name.
Line 447: “lower calorimetric peak”: have to be considered also the components ratio in the sample.
Line 449: “DSC scan of lyophilized ALP SLNs did not exhibit a sharp lower calorimetric peak”: there is a peak at about 140 °C and showed least partly crystalline state of ALP and not full amorphous. The similar can be stated from XRD pattern (line 463). It has been discussed in several papers that melting temperature of nanoparticles is dependent on the particle size.
In line 28 “molecular dispersion of crystalline alprazolam in Alp-SLNs”and In line 463 “ the molecular level and alprazolam is present in amorphous form in ALP-SLNs.” Please reconsider it!
Line 492: please, sing “n” as exponent.
In line 507: “accpatable” please, correct it.
The References need to be checked and corrected according to the Instructions:
e. g. :
Journal Articles:
1. Author 1, A.B.; Author 2, C.D. Title of the article. Abbreviated Journal Name Year, Volume, page range.
Author Response
Reviewer 2
The authors of the manuscript present the synthesis and characterization of the drug Alprazolam /solid lipid nanoparticle suspension using hot melt encapsulation technology. The influence of formulation parameters was investigated. The study is interesting and contributes to scientific relevance to the theme. The cited references give scientific support to the proposed study. The methodology consists of several studies, well-founded, and the results and discussion are presented in most cases clearly. The present work is good in quality, but I recommend the publication after a major revision, as follows:
Response: The authors are highly thankful to the reviewer for sparing his quality time and generating useful comments. These comments were very helpful in improving our manuscript and will also be considered in our future research. All these comments have been addressed point by point and highlighted in Yellow colour.
- Line 22: “PI values < 0.3 nm” PI has no unit
Response: Unit was removed.
- Introduction: there is a luck of previous studies about alprazolam SLS composites (although it is mentioned in section 4 in lines 433-442 (27. Singh, A.P., S.K. Saraf, and S.A. Saraf, SLN approach for nose-to-brain delivery of alprazolam. Drug delivery and translational research, 2012. 2(6): p. 498-507.)
Response: Previous studies have been added from lines and highlighted in the Manuscript.
- Introduction: Please, mention possible preparation methods for SLS composites. E.g. from Khairnar et al. Review on the Scale-Up Methods for the Preparation of Solid Lipid Nanoparticles. Pharmaceutics 2022, 14,1886. https://doi.org/10.3390/pharmaceutics14091886 or others. The author make it clear in the main part that the ALP-SLNs are suspension in form.
Response: Thank you for the suggestion. The mentioned article has been consulted and cited in introduction.
- In line 82: when the “current research aimed to synthesize SLNs of ALP using Compritol 888 ATO” why was lipid screening done? (Section 2.2. and 3.1.1.). If the authors want to insert Section 3.1.1. In this paper some information must be provided about possible other lipids and mentioned the results in Discussion.
Response: The content was revised and highlighted in respective sections of manuscript.
- In line 152 „…along with the standard deviation of the measurements were calculated”. Standard deviation data are not given in Table 3.
Response: The authors acknowledge the reviewer suggestion and Standard deviation has been included in Table 3.
- Section 2.9. What was the reason of the selected two pH for the drug release studies? Please, explain it in Discussion.
Response: The authors acknowledge reviewer concern and this point has been added and highlighted in the discussion.
- Section 2.11 and 2.12. : Please, put after Section 2.8 because these methods are also for physical characteristics determination (similar with Section 3.8 and 3.9.)
Response: Section 2.11, 2.12, 3.8 and 3.9 have been rearranged as per your suggestion.
- In line 226: the XRD investigation was made of “driedalprazolam formulation (ALP-SLNs)”, please, complemented it.
Response: The correction was incorporated and highlighted.
- Table 2: Because of the used surfactant concentration and the lipid:surfactant ratio were constant there is no need to put in the table. The authors can give them as non variable factors in Section 2.3. What is the drug loading without % (ratio?)?
Response: The point was addressed and Table 2 was revised as per your kind suggestion.
- In line 257: please correct it: “parameters for all formulations are listed in Table 2”.
Response: Line 257 was corrected and highlighted.
- In line 292: “The shape of SLNs was observed to be spheroid and the surface intact at various magnifications” please, correct it. Generally comment: please check where is more ALP-SLNs and where is only one sample (e. g. AF6, is it in Fig.4 as SEM or FTIR, XRD, DSC). Can be signed as AF6 and not ALP-SLNs in figures and also in the text.
Response: The suggested changes were incorporated. ALP-SLNs in figures and text have been replaced by AF6.
- In line 298, Table 4: it is not necessary to give all the detailed data of PS by SEM, the author can give only the average value in text. The Table 4. can be omitted (detailed data can be put in supplementary materials).
Response: Average particle size by SEM was mentioned in the text and Table 4 was put in supplementary materials.
- In line 310: “The Zeta potential values of the compritol-based alprazolam SLNs are highly negative 310 from −6.71 to −13.81.” The colloid dispersion can be considered stable if the zeta potential higher than ± 30 mv (e. g. Kashanian et al. https://doi.org/10.2147/IJN.S20849). But it should be noted that it is especially for electrostatic stabilized colloids. The “highly” is not appropriate for these values. Similar in line 438.
Response: Thank you for the valuable suggestion. The required correction was incorporated.
- Line 329: In Fig. 6. E is an absorbance spectrum?
Response: Fig.6 E has included transmittance pattern.
- In line 362: “…show ALP release in a sustained manner over a duration of 30 h” please, completed.
Response: The authors are thankful and now the line has been completed.
- In line 411: Please, give the BBB meaning at first appearance.
Response: BBB appeared in first appearance has been defined and highlighted.
- In lines 420-425: The amount of the lipid and ALP were given in ratio in Table 2, there is no data of 10 mg and 80 mg and %. E. g. AF1 has 5 mg lipid and 1 mg ALP. Better to give the ratio or the sample name.
Response: Thank for this technical comment. The point was revised in the manuscript and highlighted.
- Line 447: “lower calorimetric peak”: have to be considered also the components ratio in the sample.
Response: The required point has been discussed in the manuscript and content was highlighted.
- Line 449: “DSC scan of lyophilized ALP SLNs did not exhibit a sharp lower calorimetric peak”: there is a peak at about 140 °C and showed least partly crystalline state of ALP and not full amorphous. The similar can be stated from XRD pattern (line 463). It has been discussed in several papers that melting temperature of nanoparticles is dependent on the particle size.
Response: The suggested point has been included.
- In line 28 “molecular dispersion of crystalline alprazolam in Alp-SLNs”and In line 463 “ the molecular level and alprazolam is present in amorphous form in ALP-SLNs.” Please reconsider it!
Response: The highlighted point has been reconsidered.
- Line 492: please, sing “n” as exponent.
Response: “n” was mentioned as exponent.
- In line 507: “accpatable” please, correct it.
Response: The word was corrected.
- The References need to be checked and corrected according to the Instructions:
- g. :
Journal Articles:
Author 1, A.B.; Author 2, C.D. Title of the article. Abbreviated Journal Name Year, Volume, page range.
Response: The references were checked and corrected according to instructions.
Round 2
Reviewer 1 Report
I'm glad to receive your reply. The article is generally great, but there are still a few places to be revised. Because the overall structure of your article has been changed, please change the title and number of the chart, otherwise it will look very chaotic.
Author Response
Response to Reviewer 1 (Round 2)
Comments and Suggestions for Authors: I'm glad to receive your reply. The article is generally great, but there are still a few places to be revised. Because the overall structure of your article has been changed, please change the title and number of the chart, otherwise it will look very chaotic.
Response: The manuscript has been revised according to your valuable comments and suggestions. The title and number of charts have been changed. The English has been revised to enhance readability.
Reviewer 2 Report
There is no further comments.
Author Response
Response to Reviewer 2 (Round 2)
Comments and Suggestions for Authors: There is no further comments.
Response: Thank you very much for sparing your valuable time for reviewing the revised manuscript. Your suggestions have contributed a lot to enhance quality of our manuscript. English language and spelling errors have been re-checked by authors and corrected.